# Implications of Autonomous Vehicles for Accessibility and Transport Equity: A Framework Based on Literature

**Alberto Dianin** [1,2,*] **, Elisa Ravazzoli** [2] **and Georg Hauger** [1]

1     Faculty of Architecture and Spatial Planning, Vienna University of Technology, Karlsgasse 11,
      A-1040 Vienna, Austria; georg.hauger@tuwien.ac.at
2     Eurac Research, Institute for Regional Development, Viale Druso 1, I-39100 Bolzano, Italy;
      elisa.ravazzoli@eurac.edu
*     Correspondence: alberto.dianin@eurac.edu

**Abstract:** Increasing accessibility and balancing its distribution across space and social groups are two fundamental goals to make transport more sustainable and equitable. In the next decades, autonomous vehicles (AVs) could significantly transform the transport system, influencing accessibility and transport equity. In particular, depending on the assumed features of AVs (e.g., private or collective) and the considered spatial, social, and regulative context (e.g., rural or urban areas), impacts may be very different. Nevertheless, research in this field is still limited, and the relationship between AV assumptions and accessibility impacts is still partially unclear. This paper aims to provide a framework of the key and emerging aspects related to the implications of AVs for accessibility and transport equity. To set this framework, we perform an analysis of the scientific literature based on a conceptual model describing the implications of AVs for the distribution of accessibility across space and social groups. We recognize four main expected impacts of AVs on accessibility: (1) accessibility polarization, (2) accessibility sprawl, (3) exacerbation of social accessibility inequities, and (4) alleviation of social accessibility inequities. These impacts are described and analyzed in relation to the main AV assumptions expected to trigger them through different mechanisms. Based on the results, some recommendations for future studies intending to focus on the relation between AVs, accessibility, and transport equity are provided.

**Keywords:** automated transport; accessibility; equity; assumptions; impacts; literature analysis

## 1. Introduction

Accessibility, i.e., the "extent to which land-use and transport systems enable (groups of) individuals to reach activities or destinations by means of a (combination of) transport mode(s)" [1] is a key concept for transport policy. Its increase and balanced distribution across space and social groups are two key targets influencing the sustainability and fairness of the transport system [2–4]. In the next decades, autonomous vehicles (AVs) could trigger big changes in accessibility, with related implications for transport equity [5,6]. For instance, the use of expensive private AVs could initially be more beneficial for the accessibility of high-income groups, but out of reach for people with a low income who cannot afford them. At the same time, massive investments in ride-sourcing services to the detriment of public transport (PT) could increase accessibility in high-demand areas, but be harmful to low-demand zones [7,8]. On the other hand, the development of affordable ride-sourcing schemes combined with improved PT could increase the accessibility of social groups relying on PT, as well as provide low-demand areas with more competitive services (e.g., [9,10]). In fact, this is the well-known social and spatial diffusion model.

Despite these opportunities and risks, the implications of AVs for accessibility and transport equity are still little explored [11], although the reduction of inequalities and increase of accessibility are among the top priorities for European and international sustainable development [12]. Therefore, it is of great importance to better understand how

a disruptive innovation such as transport automation may influence these aspects. So far, most of the literature has focused on the mere technical challenges related to the introduction of AVs [13]. In contrast, a minor but increasing number of contributions discuss the impacts on the transport system, behaviors, and land use [14–16]. Among them, various contributions include the concept of accessibility to forecast the effects of AVs on modal choices and travel patterns, or the expected attractiveness of certain areas and the relocation patterns AVs may trigger (e.g., [17–19]). Nevertheless, only a few studies focus on accessibility as the main research topic (e.g., [5,20]). Therefore, accessibility is a key concept used in transport literature to shed light on many changes that AVs may trigger, but it is not frequently taken as a core topic of discussion.

This paper aims to set a framework of the key and emerging aspects related to the implications of AVs for accessibility and transport equity. To set this framework, an analysis of the scientific literature is performed. This analysis is based on a conceptual model, which describes how AVs may influence the distribution of accessibility across space and social groups. By adopting this model and interpreting the literature accordingly, four main potential impacts of AVs on accessibility are identified, and the main assumptions that are expected to trigger them are discussed. Therefore, through this process, the paper extrapolates the main concepts, assumptions, and impacts from the literature and explores how they are linked with each other. By developing this study, we aim to provide two main contributions compared to other literature analyses in the context of AVs. First, we provide an overview of an under-researched aspect of the social impacts of AVs; second, we point out the correlation between impacts and assumptions. Future studies intending to explore the relation between AVs, accessibility, and transport equity can use this work as a starting knowledge framework.

The paper is organized as follows. Section 2 explains how this study places AVs in the scientific debate around accessibility and transport equity. Afterward, it presents the conceptual model used for the analysis, the selected literature, and the criteria used for the selection. According to the conceptual model, Section 3 identifies and describes four main accessibility impacts of AVs and analyzes the key AV assumptions that are expected to trigger them. Finally, Section 4 discusses the investigation results, while Section 5 concludes the article by providing some recommendations for future studies intending to focus on the relation between AVs, accessibility, and transport equity.

## 2. Methodology

After placing this study in the scientific debate about accessibility and transport equity (Section 2.1), this section presents the conceptual model used for the analysis (Section 2.2) and the selected literature (Section 2.3). Based on these two last elements, the fundamental research is then performed in Section 3.

### 2.1. Framing AVs in the Accessibility and Transport Equity Debate

As stated by [21], considerable academic work has gone into the development of accessibility concepts and measures, starting with notable writings in the 1950s such as the well-known work by [22], who defined accessibility as the "potential of opportunities for interactions". Drawing on this concept, many scholars set accessibility analyses, while others proposed definitions/conceptualizations explaining how accessibility can be interpreted (e.g., [1,23,24]). In particular, [1] defined accessibility as the "extent to which land-use and transport systems enable (groups of) individuals to reach activities or destinations by means of a (combination of) transport mode(s)". As such, accessibility is influenced by four main elements, called "accessibility components": the land use and transport systems, and the individual and temporal possibilities/constraints (explained in detail in Section 2.2). Modifications in one or more of these components and their interactions trigger changes in the accessibility level. On this basis, many studies deployed different indicators and models to point out how the introduction of varying transport and land use policies may change accessibility (see, e.g., the comprehensive reviews by [25]).

The increasing relevance of accessibility in transport research also increased its significance as a parameter to measure transport equity, which can be defined as the "morally proper distribution of (transport) benefits and costs over members of society" [26]. As stated in the review by [27], the equity of accessibility distribution is the most evaluated topic in the transport equity field. According to [4], the relationship between equity and accessibility evaluation involves two main aspects: the fair distribution of accessibility benefits across space (e.g., different areas or regions) and across individuals or groups of the society (e.g., people with high and low income). These two perspectives have great political relevance, especially when they highlight how specific transport policies could increase the accessibility disparities among regions or income classes. In this respect, making accessibility distribution more balanced becomes as important as improving its overall level.

Within this accessibility and transport equity framework, AVs are one of the main innovations that are expected to produce significant changes, similarly to, e.g., information and communication technologies (ICTs, e.g., [28]) or high-speed railway lines (HSRs, e.g., [29]). Nevertheless, studies explicitly focusing on the relation AV–accessibility–equity are still limited, and this paper aims to collect them under a single framework.

### 2.2. Conceptual Model

According to this scientific framework, we propose a conceptual model that describes how AVs may influence the distribution of accessibility across space and social groups and thus affect transport equity (Figure 1). This model is based on the concept of the four accessibility components by [1], as well as on the conceptualization of the relation between equity and accessibility evaluation provided by [4]. Additionally, it takes into account the model of the accessibility impacts of AVs on social inclusion by [6] (see the conceptual model of the long-term implications of automated vehicles for social inclusion).

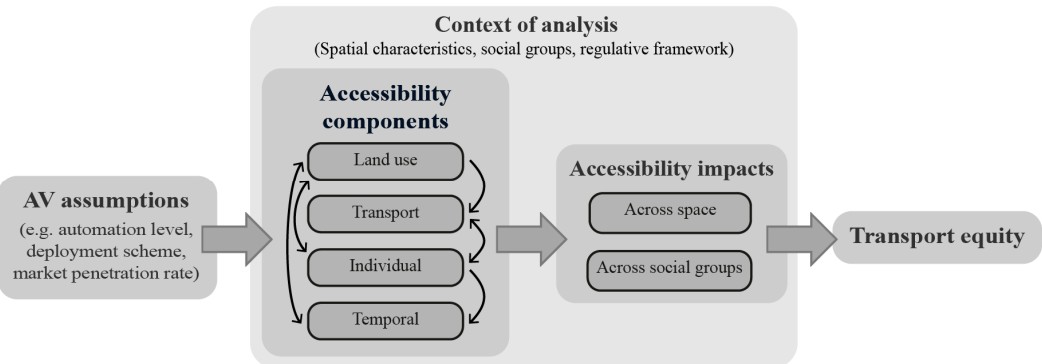

**Figure 1.** Conceptual model. A conceptual model describing how AVs may influence the distribution of accessibility across space and social groups and thus affect transport equity.

As is visible in Figure 1, the model is made up of four elements: the AV assumptions expected to transform the transport system, the four accessibility components influenced by AVs, the expected accessibility impacts across space and social groups, and the context of analysis describing variations in the characteristics of the accessibility components influencing the impacts. Depending on the AV assumptions (e.g., private vs. collective AVs) and the context of analysis (e.g., urban vs. rural area, young vs. older adults), the influence on the four accessibility components and the related accessibility impacts may be different. The elements composing the model and their interconnections are further explained below.

AV assumptions: They comprise several features of AVs, such as their level of technological development (e.g., the classification in SAE Levels 0–5; [30]), their deployment scheme (private, collective, or shared), their cooperation in a coordinated system, or their expected market penetration in the vehicular fleet. Depending on the considered AV assumptions, the four accessibility components may be affected in multiple ways. For example, the spread of fully automated ride-sourcing systems could decrease the vehicular fleet needed to cover the demand (transport component), the need for on-street parking spots (land use component), the physical limits of the elderly (individual component), and the time travelers spend to find a parking spot (temporal component).

Accessibility components: According to [1], the land use component concerns the availability and location of opportunities and demand for opportunities (such as workplaces and households). The transport component includes the features of the transport supply and of the passenger or freight travel (such as road capacity and modal split). The individual component comprises possibilities and constraints that characterize individuals such as income, physical/mental condition, or car ownership. Finally, the temporal component describes time-related opportunities and constraints conditioning the individuals and the opportunities, such as the daily time at their disposal for grocery shopping and the opening hours of groceries. By influencing one or more of these components, AVs change accessibility. Additionally, the components influence each other through indirect relationships (Figure 1, [1]).

Accessibility impacts: Two kinds of accessibility distribution impacts are considered in the model: those across space and social groups. Together, these two distribution effects influence the overall equity of the transport system [4]. In particular, changes in the distribution of accessibility across space affect the "horizontal" or spatial equity, i.e., the fairness of the distribution of benefits among different areas (such as urban and rural). In contrast, changes in the distribution of accessibility across social groups affect the "vertical" or social equity, i.e., the fairness of the distribution of benefits among different social groups (such as wealthy and poor) (we refer to [2,31] for details on the concepts of "horizontal" and "vertical" equity).

Context of analysis: This comprises the spatial characteristics (e.g., rural or urban areas), social groups (e.g., young or older adults), and regulative framework (e.g., spatial and transport planning laws) shaping the context where accessibility is analyzed. These three elements do not exert a direct influence on the accessibility components; rather, they describe variations of their characteristics. For instance, a retired person is likely to have a different physical condition than a younger worker (individual component), as well as a different amount of time at their disposal for discretionary activities (temporal component). Yet, a rural area is likely to have a smaller and more dispersed number of workplaces than an urban area (land use component). Depending on such variations, the same AV assumptions may lead to different accessibility impacts.

### 2.3. Literature Selected for the Analysis

Scientific studies focusing on the impacts of AVs on accessibility are not numerous. However, several works include accessibility in their wider analysis in order to (a) forecast transport demand impacts, (b) forecast land use impacts, (c) discuss social impacts, and (d) discuss a broad set of impacts such as economic and environmental ones. As is visible in Table 1, the 41 contributions selected for this study cover this range of topics. Some of them adopt quantitative modeling methods, while others use non-modeling approaches.

**Table 1.** Selected literature. Overview of selected contributions broken down by their main topic and methodological approach.

| Main Topic | Methodological Approach | |
|---|---|---|
| | Quantitative modeling methods | Non-modeling approaches |
| Accessibility impacts of AVs | **Childress et al., 2015 [1]; Meyer et al., 2017; Luo et al., 2019; Nahmias-Biran et al., 2020** | **Milakis et al., 2018; Papa and Ferreira, 2018** |
| | Quantitative modeling methods | |
| Transport demand impacts of AVs (including accessibility) | Kim et al., 2015; Azevedo et al., 2016; Liu et al., 2017; Basu et al., 2018; Nahmias-Biran et al., 2019; Vyas et al., 2019; Coppola and Silvestri, 2019; Le et al., 2019; **Basu and Ferreira, 2020** | |
| | Quantitative modeling methods | |
| Land use impacts of AVs (including accessibility) | **Thakur et al., 2016**; Zhang, 2017; Zhang and Guhathakurta, 2018; **Gelauff et al., 2019**; **Kang and Kim, 2019**; May et al., 2020; **Basu and Ferreira, 2020a; Basu and Ferreira, 2020b**; Kim et al., 2020 | |
| | Quantitative modeling methods | Non-modeling approaches |
| Social impacts of AVs (including accessibility) | **Harper et al., 2016; Cohn et al., 2019** | Brown and Taylor, 2018; Kuzio, 2019; Fitt et al., 2019; Pudane, et al., 2019; **Cohen et al., 2020**; Singleton et al., 2020; Faber and van Lierop, 2020; **Milakis and van Wee, 2020**; Sparrow and Howard, 2020; Shirgaokar, 2020 |
| | Quantitative modeling methods | Non-modeling approaches |
| Broad set of impacts of AVs (including accessibility) | Martinez and Viegas, 2017; Abe, 2019 | Ticoll, 2015; Sessa et al., 2016; González-González et al., 2019 |

[1] Articles for which backward and forward snowballing have been used are highlighted in bold.

These contributions were first collected in winter 2020 and then in spring 2021 from Scopus, Web of Science, and Google Scholar. Only peer-reviewed articles and conference papers written in English are considered, while project and industrial reports are excluded in order to focus on documents that have received a similar quality control process (peer review). Contributions are selected independently of their geographical area and publication period. Nevertheless, all selected contributions were published between 2015 and 2020 (70% after 2018). This figure is consistent with the findings of [11,13], who highlight that the interest in the broad social impacts of AVs started growing after 2015. The keywords "autonomous vehicles" and "accessibility" have been jointly used through the Boolean operator AND. To find papers dealing with the connection between AVs, accessibility, and transport equity, the keyword "equity" has also been used together with the previous ones. The studies including elements concerning accessibility but not dealing with the accessibility impacts of AVs have been discarded. In contrast, those exploring the accessibility and transport equity impacts of AVs with the greatest level of detail have been used for backward and forward snowballing. Alternative keywords such as "automated vehicles", "self-driving cars", and "self-driving vehicles" have been used in all operations.

## 3. Accessibility Impacts of AVs across Space and Social Groups

In this section, we focus on two specific aspects included in the conceptual model in order to analyze the selected contributions, the accessibility impacts and the related AV assumptions that are expected to trigger them, by influencing the four accessibility components.

Generally, there is no wide consensus on the expected impacts of AVs on accessibility; rather, results are in some cases contradictory (e.g., [6,20,32–35]). This depends on the great heterogeneity of AV assumptions, spatial contexts, and social groups considered, as well on the several uncertainties about AVs. Nevertheless, it is still possible to summarize four main accessibility impacts, which derive from a subset of seven more specific impacts found in the literature (Figure 2, see sample references). These four main impacts are (1) accessibility

polarization driven by an increase of accessibility mainly in central areas; (2) accessibility sprawl due to an increase of accessibility mainly in suburban and even rural areas; (3) exacerbation of social accessibility inequities due to the highest accessibility benefits for the most appealing market sectors; and (4) alleviation of social accessibility inequities thanks to a more balanced distribution of accessibility across different social groups.

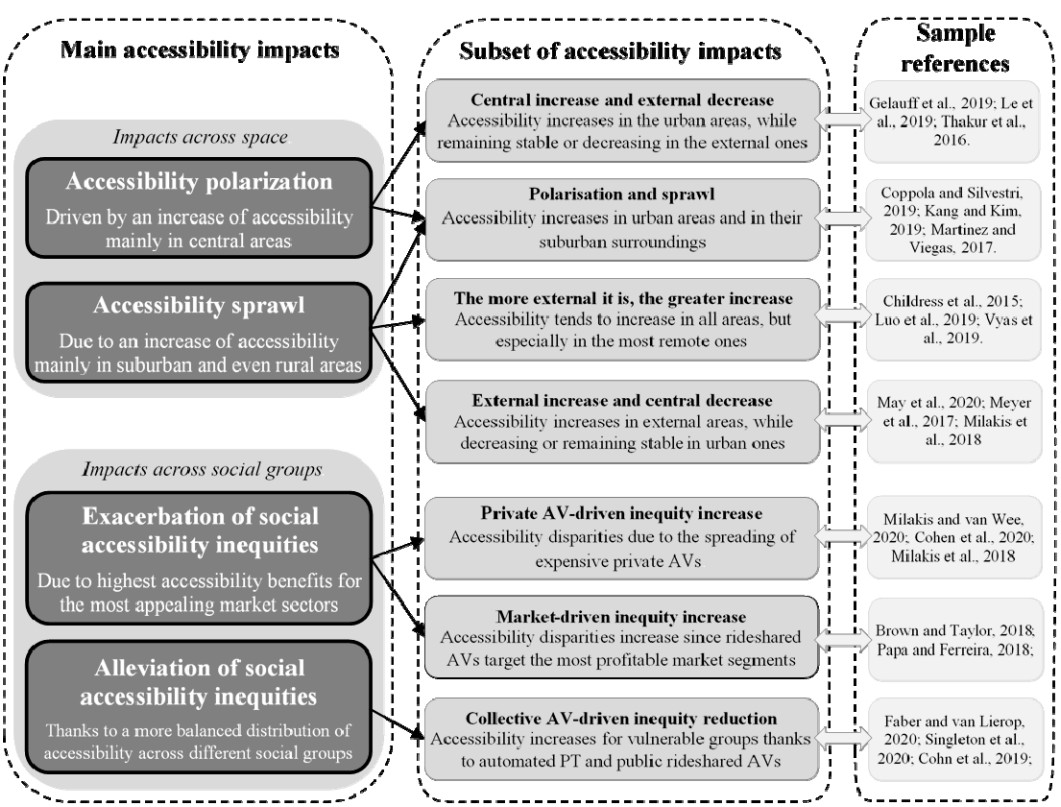

**Figure 2.** Main accessibility impacts. The four main expected accessibility impacts and the related subset.

The next four Sections 3.1–3.4 discuss these four main accessibility impacts and point out which ones are the key AV assumptions triggering them and through which mechanism. Afterward, Section 3.5 presents the results in a summary table.

### 3.1. Accessibility Polarization

3.1.1. Description of the Accessibility Impact

In urban centers, accessibility could significantly increase due to the spread of AVs. This in turn could lead to the concentration of inhabitants and activities in the most urbanized areas. For instance, ref. [36] found out that the central commercial area in Seoul is expected to get the greatest accessibility benefits after the introduction of AVs, while most of the land usage in its surroundings will change to residential land use. A similar trend is highlighted in [18]: two scenarios developed in their study show that inhabitants could cluster in urban areas and especially in the biggest and most attractive cities where accessibility is expected to increase most. In quantitative terms, AVs could trigger an increase by ca 10% of the population in the most urbanized areas by 2050. Even the analysis by [37] suggests that, under specific conditions, the inner zone of Melbourne could experience high accessibility benefits to the detriment of outer suburbs. Apart from these studies mainly adopting Land Use Transport Interaction Models (LUTIs), other types of contributions also highlight similar trends. For instance, ref. [20] apply a Q-method to a sample of seventeen international accessibility experts who think that AVs could trigger the densification of consolidated city centers in parallel with the growth of new minor

centers. This trend is expected because of the accessibility gains that AVs may provide in urban areas.

### 3.1.2. Main AV Assumptions Linked to the Impact

Ride-sourcing AV systems and automated PT services are expected to play a key role in accessibility polarization (e.g., [17,38,39]). For instance, ref. [18] show that the introduction of shared AVs as new door-to-door mobility services and the parallel development of automated PT make urban trips faster and cheaper, as well as the access, egress, and waiting times lower. This influences the modal split, since a strong increase in the usage of shared and collective transport means both at the national level and especially in the largest cities is expected. At the same time, the distribution of households could change, with many people relocating to the most urbanized areas. Consistent results are provided by [17], exploring the impacts of different AV scenarios in the metropolitan area of Rome. Even in this case, shared services feeding the mass rapid transit stations and thus increasing the PT network's attractiveness are crucial to increasing accessibility in the urban core and limiting urban sprawl.

Combining such ride-sourcing AV systems with management measures such as car use restrictions, car-free areas, and road-pricing schemes may provide the highest urban accessibility benefits [17,18,40,41]. These measures are likely to be needed to manage future challenges brought by automation and prevent a modal shift of users from PT to shared AVs in the urban areas at risk of congestion [42]. Accordingly, ref. [41] introduced a ban to the circulation of motorized private modes in the central business district in one of their scenarios. Conversely, busses and mass rapid transit were allowed to circulate. This lead to positive accessibility outcomes at an urban level. The same applies to [18], who assumed the introduction of road pricing schemes compensating the expected loss of parking fees in urban areas. Additionally, they considered that operators managing AV fleets will make trip fares time-varying, with higher fares during peak hours.

Another important assumption that is often coupled with ride-sourcing systems deployment is the full automation and high market penetration of AVs (e.g., [43–47]). Indeed, the full automation (SAE Level 5; [30]) is expected to optimize the operation of sharing schemes based on AVs that pick-up and drop-off users on-demand and manage to park autonomously. This is well highlighted by [18]: the scenarios assuming full automation make the accessibility gains of urban areas much more evident than partial automation scenarios and the number of households expected to move in urban areas significantly higher.

Regarding the perception of travel time disutility, various studies observe that assuming only partial reduction provides the highest accessibility benefits in central areas (e.g., [37,47]). Indeed, the higher the reduction of perceived travel time disutility, the higher the expected willingness of users to travel longer distances and the perceived accessibility of suburban and rural areas. For instance, ref. [37] find out that when AVs are used for ride-sourcing services and there is no reduction in the travel time disutility, this leads to the highest accessibility increase in the urban core of Melbourne. In contrast, assuming a reduction of travel time disutility of 50% coupled with the same ride-sourcing scheme could generate more diffused accessibility gains in outer suburbs.

### 3.2. *Accessibility Sprawl*
### 3.2.1. Description of the Accessibility Impact

In this case, suburban and even rural areas could live the greatest accessibility gains and thus stimulate a new urban sprawl phase, with several households relocating to outer areas and new residential outskirts. Various contributions discuss this potential scenario. For instance, ref. [5] use the Swiss national transport model to simulate the accessibility impacts of AV on the whole of Switzerland at the municipal level. Results indicate that, under specific circumstances, the already well-connected suburban and rural municipalities experience the strongest accessibility increase (up to +76% in the most extreme scenario),

whilst the effects on urban areas are almost negligible or even negative. In contrast, the most remote rural areas register very partial accessibility benefits. These results are in line with those provided by previous works such as those of [48,49]. The former in particular indicates that accessibility is expected to increase in the whole Seattle region; however, the highest gains are registered for the households living in the most remote rural areas, where increases between +106% and +260% could be registered. Also in later works (as [50,51]), accessibility sprawl is often expected. In particular, ref. [50] apply the large-scale agent-based disaggregate simulation model MATSim to the Gunma prefecture in Japan. Results show that, overall, an accessibility increase between +23.2% and +36.3% is expected, with the highest gains registered in the particularly remote areas, which could become areas of new suburbanization in the next decades.

3.2.2. Main AV Assumptions Linked to the Impact

Private AVs are by far the most important factor expected to increase accessibility in suburban and rural areas (e.g., [5,50,51]). By improving current road capacity and the perception of on-board travel time compared to human-driven cars (two features discussed below), private AVs are expected to affect the travel time and the comfort of the travel, stimulating people to travel longer distances as well as to relocate to more dispersed residential locations in the long term. The key role of private AVs for accessibility sprawl is well highlighted in the set of scenarios developed by [50]. In scenarios 1–4, a progressively higher diffusion of private AVs is assumed, with the percentage of users owning private AVs varying from 10% (Scenario 1) to 50% (Scenario 4). Across scenarios, the accessibility benefits affecting the urban area remain always limited. In contrast, rural areas register a progressive growth of accessibility gains in parallel with a higher diffusion of private AVs. Besides private AVs, the spread of shared AVs with low fares could increase accessibility in suburban and rural areas (e.g., [9,50,52]). In these contexts, people are used to traveling long distances to reach workplaces and amenities; therefore, the impacts of cheap shared AVs on the cost of rural travels are expected to be higher than for urban travels [48,51].

As anticipated, the diffusion of private AVs has a disruptive effect according to the assumed effects on road capacity and travel time disutility (e.g., [51,53–56]). Indeed, road capacity directly affects travel time and the risk of congestion on the road network, while travel time disutility affects the perception of the time spent on-board and therefore the willingness of users to travel longer or shorter distances. As regards road capacity, ref. [5] highlight how different assumptions can lead to very different accessibility implications. In their scenarios, the authors assume a capacity increase for the extra-urban network of either +80% (conservative increase) or +270% (optimistic increase), while the value of the urban network is fixed at +40%. When considering the conservative increase, the urban areas (such as around Zurich and Bern) are found to experience accessibility losses, while suburban and rural municipalities register slight increases. In contrast, when considering the optimistic increase, considerable accessibility gains are registered in almost all suburban and rural municipalities. This suggests how road capacity gains may play a key role. However, there are many uncertainties about the expected market penetration rate of AVs, the interaction between human-driven and self-driving cars, and even about the cooperation among AVs, which make road capacity gains challenging to forecast [47]. As for the reduction of the perceived travel time disutility, very heterogeneous assumptions are available (e.g., [37,51,54,55]). Ref. [51] compare two scenarios where a reduction by 25% and 50% in the travel time disutility is assumed. Results show that the assumption of a 50% decrease has a relatively stronger impact on accessibility, especially for rural and suburban areas.

Other relevant elements leading to accessibility sprawl regard the assumed reaction of the demand to the introduction of private and shared AVs in terms of induced demand and modal shift. Various studies assume that the benefits provided by AVs for travel time, comfort, and monetary costs (in the case of shared AVs) could encourage more people to travel and make more extended travels. Moreover, there will be many empty rides given by the operation of shared AVs. Due to such induced demand, the benefits of AVs could be offset in urban areas, where traffic and congestion issues are more significant (e.g., [5,48]). Conversely, suburban and rural areas are expected to obtain accessibility benefits anyway, since their road network is not subject to such high pressure. As regards the modal shift, some contributions assume that many commuters will shift from PT to shared AVs, given the higher flexibility and the competitive fares (e.g., [5,19,20]). Even in this case, this modal shift could generate relevant issues and accessibility losses in urban areas where PT also plays a crucial role in mitigating congestion. The same does not apply to rural areas, where this modal shift is not expected to offset the accessibility benefits.

### 3.3. Exacerbation of Social Accessibility Inequities

### 3.3.1. Description of the Accessibility Impact

As regards the accessibility impacts of AVs across social groups, several studies highlight the risk that, under specific circumstances, AVs could increase social inequity in different ways (e.g., [6,35,57–59]). In particular, people who cannot afford a private vehicle or to spend much money on transport services could be excluded from the benefits of transport automation, which will be mainly accessible for people with high income (e.g., [20,60]). This is one of the main social equity concerns about AVs, as highlighted by [20,52]. Other inequity issues could affect users who do not represent profitable market segments for private providers of transport services performed with AVs. For instance, ref. [34] highlight that vulnerable social groups such as the elderly, people living in low-demand areas, or people seeking transport services during off-peak hours could get minimal advantage from AVs if they operate just to maximize the profit. Another group that could be negatively affected is people who rely on PT to reach their destinations. According to [6,34], they could experience relevant accessibility losses if the spread of automation causes a reduction of traditional PT services. Finally, even the users who are not familiar with or cannot access online booking tools and payment methods are expected to experience substantial transport inequities in a future transport system based on AVs [14,39].

### 3.3.2. Main AV Assumptions Linked to the Impact

The transition phase in the development of AVs (partial automation) is considered a relevant factor that could exacerbate accessibility inequities between people with low and high income (e.g., [6,60–62]). Ref. [52] highlight that, in the initial stages of implementation, AVs will be unaffordable for a large part of the population, while only people with high income will be able to buy this kind of new technology. As such, only people with high income will experience the accessibility improvements concerning comfort and perception of travel time disutility. The phase of partial automation might also exacerbate the accessibility inequities between people with and without a driving license, since not fully automated vehicles will be not able to drive autonomously in all conditions, and the passenger could be asked to take over the ambitious tasks on some occasions [30].

These issues could be strongly exacerbated if combined with an ownership-based paradigm dominating the transport sector also after a vast deployment of AVs (e.g., [35,39,57]). The wide spread of ride-sourcing systems or shared AVs may be beneficial for people who could use this technology without incurring the capital costs related to the ownership (covered by the service provider). However, if AVs will be mainly deployed as private means, then people who cannot bear high capital costs will be excluded from the accessibility benefits provided by this technology. Additionally, the expenses related to the ownership of traditional human-driven cars could also get higher, since, for example, insurance policies

will be more expensive. This will mainly affect people who cannot afford to buy an AV but need a car.

However, even the deployment of ride-sourcing services using AVs could lead to controversial effects. [34] show that inequities could increase if ride-sourcing systems will be managed by private companies acting to maximize profitability rather than to provide a public service. In this case, users who do not represent an appealing market sector (such as people traveling during off-peak hours or living in low-demand areas) could be excluded and have no access to such transport options. As such, the already existing accessibility inequities could be exacerbated. At the same time, the high spread of shared mobility could undermine, in some cases, the provision of traditional PT, with negative consequences for those relying on this type of service. On this point, there is no broad consensus. According to [6], this could be likely especially for people living in suburban and rural areas, as well as for those living in areas experiencing suburbanization processes due to the spread of AVs. In contrast, various experts interviewed in [20] believe that the spread of AVs will not harm public investments in PT.

Another important AV assumption that can generate social inequities is the total digitalization of booking and purchasing operations related to transport services [14,39]. Ride-sourcing services are supposed to work through apps allowing the booking of the ride and the payment via credit card. However, people who are not familiar with these tools or even do not own them (e.g., due to budget constraints) will have relevant difficulties in using such transport services. According to the data presented by [34], this issue is not negligible, since low-income Americans without a checking or savings account represented ca 7% of US households in 2015. To conclude, AVs could generate social inequities also according to the different possibilities of user groups to perform activities while traveling (e.g., [52,63]). For example, people making on-site works could hardly benefit from the possibility of working on-board. As such, AVs may increase the transport inequity among the individuals who do or do not have flexible working times, tasks, and contracts.

### 3.4. Alleviation of Social Accessibility Inequities

3.4.1. Description of the Accessibility Impact

In an optimistic vision, AVs are expected to alleviate the accessibility inequities among social groups by increasing, in particular, the accessibility of vulnerable groups such as the elderly, children, and people with low income as well as with physical and sensory disabilities (e.g., [54,57,58,64,65]). All these groups are characterized by three main limitations, which affect their accessibility: the impossibility of driving, the impossibility of owning a car, and the consequent dependency on collective transport means. AVs may be beneficial in multiple ways. For example, AVs may allow users unable to drive to benefit from the advantages of private transport (such as short travel times, door-to-door linkages, and comfort). In this respect, studies such as those of [5,61] have discussed how and to what extent new users such as the elderly could enter the private transport market, affect the demand along the road network, and thus influence the overall accessibility. At the same time, other works such as those of [35,58] have highlighted that AVs could allow people not able to drive to engage in more activities, increasing their accessibility and decreasing their risk for social exclusion. At the same time, AVs could lower the accessibility inequities related to the ownership of a vehicle. Indeed, people with low income who cannot bear the high capital costs associated with car ownership could use AVs anyway thanks to the introduction of shared and pooled services (to be provided with competitive fares). Moreover, the expected ability of these services to provide door-to-door connections could provide great support for the accessibility of people affected by physical or sensory disabilities, although specific design and operational adjustments would be necessary [6].

### 3.4.2. Main AV Assumptions Linked to the Impact

The achievement of a full automation (SAE Level 5; [30]) is a first requirement to make AVs able to decrease the social inequities existing between people able and unable to drive. This is well highlighted in the study by [6]. Based on a conceptual model, the authors highlight that SAE Level 3 AVs are not expected to provide any relevant accessibility gains to the elderly and people with a physical and sensory disability. Conversely, SAE Level 4 and 5 AVs should allow them to engage in more activities, decrease travel time, and have more flexible transport options at their disposal, resulting in relevant accessibility gains. A high level of automation could be also useful to reduce the time constraints that affect some user groups, such as parents who have to take their children to school and adjust their activities and modal choices accordingly. However, it is important to highlight that all these users will also have to face higher costs for the purchase and maintenance of sophisticated vehicles. Additionally, the induced demand generated by the entrance into the private transport market of many new users (such as the elderly or pupils) could have controversial impacts on congestion and thus on accessibility (e.g., [5]).

To prevent these side effects and optimize the equity gains, fully automated vehicles should be deployed to provide affordable car-sharing and ride-sourcing services. This combination would allow decreasing the costs of travel compared to private AVs while keeping several advantages concerning travel time and flexibility. As such, this option could also be highly beneficial for people who, for budget reasons, cannot afford to own a vehicle. The equity benefits linked to this combination are confirmed, e.g., by [54], who focus on the equity impacts of AVs. The authors propose eight scenarios. Some of them assume the presence of single-occupancy AVs, while others that of high-occupancy AVs. Moreover, some scenarios also assume the reduction or enhancement of PT routes. Results reveal that the scenarios with high-occupancy AVs are expected to provide the greatest job accessibility gains to the disadvantaged population living in the Washington D.C. area. Moreover, the enhancement of traditional PT may provide an extra gain. This suggests that the integration of ride-sourcing AVs with automated PT may also play a positive role in the alleviation of existing inequities affecting users who cannot drive or own a car [17,34,39]. The "alternative scenarios" developed by [17] confirm this. The increase of PT and shared AV services tested in the area of Rome is particularly beneficial for the accessibility of people with low–medium income, who are expected to relocate (in the long run) to the areas offering this kind of accessibility benefit.

Finally, the vehicle and street design as well as the availability of multiple booking and purchasing means (not only digital) are also relevant AV assumptions to decrease inequities and facilitate the inclusion of vulnerable social groups. For example, refs. [6,34] highlight that appropriate boarding spots need to be identified to make ride-sourcing AV services also effective for people with physical or sensory disabilities. At the same time, the in-vehicle design should be adjusted. Both these aspects could lead to higher costs for these user groups, but also greater possibility of travel and engaging in activities. Yet, telephone booking systems and prepaid cards may facilitate the inclusion of people with low income or those not familiar with digital tools.

### 3.5. Summary Table

According to the previous Sections, Table 2 summarizes the main AV assumptions that are linked to the four main accessibility impacts identified in this paper (accessibility polarization and sprawl, and exacerbation and alleviation of social accessibility inequities). Additionally, it points out the main implications of such assumptions for the four accessibility components.

**Table 2.** Summary table. Main AV assumptions linked to the four accessibility impacts and their implications for the components.

| Main Accessibility Impacts across Space | | Main Accessibility Impacts across Social Groups | |
| --- | --- | --- | --- |
| Accessibility Polarization (Section 3.1) | Accessibility Sprawl (Section 3.2) | Exacerbation of Social Accessibility Inequities (Section 3.3) | Alleviation of Social Accessibility Inequities (Section 3.4) |
| **Main AV assumptions linked to the impacts** | | | |
| ■ Diffusion of ride-sourcing AV systems<br>■ Diffusion of automated PT services<br>■ Management measures such as car use restrictions, car-free areas, and road-pricing schemes<br>■ Full technical automation of AVs<br>■ High market penetration of AVs<br>■ Partial reduction of the perceived travel time disutility | ■ Diffusion of mainly private AVs<br>■ Diffusion of shared AVs with low fares<br>■ High increase of the road capacity, especially along extra-urban roads<br>■ Strong reduction of the perceived travel time disutility<br>■ Assumed induced demand due to more numerous and longer travels<br>■ Assumed modal shift from collective to shared modes | ■ Partial automation of vehicles during the transition phase<br>■ Diffusion of mainly private AVs (ownership-based paradigm)<br>■ Establishment of ride-sourcing systems operating to maximize profitability (rather than to provide a public service)<br>■ High spread of shared mobility, to the detriment of traditional PT provision<br>■ Total digitalization of booking and purchasing operations | ■ Full automation of vehicles (able to perform all driving tasks in all situations)<br>■ High diffusion of public ride-sourcing and car-sharing systems with affordable fares<br>■ Integration of ride-sourcing and car-sharing systems with improved automated PT<br>■ Vehicle and street design making ride-sourcing and car-sharing systems suitable for all user groups<br>■ Availability of also off-line booking systems and prepaid payment methods |
| **Main implications for the accessibility components (land use, transport, individual, temporal)** | | | |
| *Land use and transport:*<br>■ Faster and cheaper urban trips by collective/shared modes<br>■ Lower access, egress, and waiting times<br>■ Lower parking-related constraints<br>■ Encouraged modal shift from private to collective/ shared modes<br>■ Discouraged modal shift from collective to shared modes<br>■ Users are discouraged from traveling longer distances<br>■ Households and activities tend to relocate to urban centers or high-density agglomerations | *Land use and transport:*<br>■ Reduced travel time by private modes<br>■ Strongly reduced monetary costs for shared mobility<br>■ Increased comfort of private modes<br>■ Increased availability of users to travel longer distances<br>■ Increased congestion on the urban road network<br>■ Households tend to relocate to more external and (cheaper) areas | *Transport, individual and temporal:*<br>■ Impossibility of buying a private AV (for people with low income)<br>■ Impossibility of riding in a partially automated AV (for people without a driving license)<br>■ Difficulty to access ride-sourcing systems (for people not representing a profitable market segment)<br>■ Higher insurance costs for people that need a car but cannot afford an AV<br>■ Difficulty to book and pay AV-based services (for people with a low digitalization rate)<br>■ Lower utility of travel time in AVs (e.g., for people who cannot work on-board) | *Transport, individual and temporal:*<br>■ Ease of access and use of ride-sourcing and car-sharing services deploying AVs (for all social groups)<br>■ Possibility of engaging in more activities (especially for people who cannot drive)<br>■ Reduced travel time and increased flexibility (for all social groups)<br>■ Ease of booking and paying AV-based services also off-line and without credit cards (for people with low digitalization rate)<br>■ Decreased time constraints related to mandatory activities (e.g., for parents accompanying children to school) |

## 4. Discussion

Through the conceptual model presented in Section 2 and the analysis performed in Section 3, this paper aims to set a framework for a better understanding of the main implications of AVs for accessibility and transport equity. In this respect, we identified four main potential accessibility impacts across space and social groups and explained which AV assumptions are expected to be crucial to generate such impacts.

According to this analysis, some general considerations can be drawn. First, literature mainly includes accessibility evaluations to discuss other phenomena such as the expected transport demand changes and household relocation patterns. Since accessibility is a key principle to provide an effective and sustainable transport system [3,21], more work on the specific relation between AVs and accessibility is needed in the upcoming years. Second, the accessibility impacts of AVs and their implications for transport equity may be very diverse depending on the considered assumptions and the observed spatial, social, and

regulative context. Indeed, AVs may both exacerbate and alleviate accessibility inequities across social groups (e.g., people with high and low income) and space (e.g., urban and rural areas). In this respect, some studies show how diverse the accessibility impacts of AVs may be (e.g., [6,18,20]). This kind of work is particularly important to shed light on the spatial and social equity implications of AVs. Third, some AV assumptions are expected to play a ground-breaking role in accessibility and transport equity, particularly the partial or full ability of AVs to take over all driving tasks, the maintenance or abandoning of an ownership-based mobility paradigm, the diffusion or not of ride-sourcing systems, the operation of ride-sourcing systems according to principles of maximal profitability or public service provision, the fare system applied to ride-sourcing systems (e.g., very expensive, very cheap, or demand sensitive), and the actual extent of road capacity and travel time perception benefits. A better understanding of all these aspects and their interlinkages in terms of accessibility is crucial.

## 5. Recommendations and Conclusions

Given this framework, it is possible to suggest some recommendations for future studies aimed at exploring the relation between AVs, accessibility, and transport equity. Some of these recommendations concern the considered AV assumptions and the context of analysis (A); others regard the approaches used for accessibility analysis (B).

(A) AV assumptions and context of analysis:

- Questioning the key AV assumptions influencing accessibility. As highlighted in Section 3, the abandoning of an ownership-based mobility paradigm and the strong diffusion of ride-sourcing systems deploying AVs are among the decisive assumptions in many studies. The spread of sharing concepts in the active mobility sector suggest that a transition in this direction is occurring (e.g., [66]). However, other data show also that big limits still exist. For instance, the National Household Travel Survey conducted in 2018 in the U.S. indicates that the average light vehicle occupancy rate in 2017 was 1.67 [67]. This value is the same as in 2009 and slightly higher than in 1995 (1.59). Therefore, in the last 20 years, the habits of Americans as concerns sharing car trips have remained almost unchanged. A similar trend is visible in Europe, where the car occupancy rate increased from 1.45 in 1995 to 1.70 in 2014 [68]. Considering this example, it is essential to question several AV assumptions according to past and current trends and by considering the policies that could encourage a future change. Various studies have conducted similar investigations, especially to understand the expected utility of on-board travel time in AVs (e.g., [69]). Extending this type of work to other key assumptions discussed in this paper and linking them to accessibility would help to better understand how AVs might actually affect accessibility.

- Enlarging the set of considered spatial, social, and regulative characteristics shaping the context of analysis. As displayed in the conceptual model (Section 2.2), the spatial, social, and regulative context of analysis defines changes in the accessibility components. Therefore, the same AV assumption may have very different consequences depending on the context we refer to. Some studies have considered this aspect by comparing the impacts of AVs across different contexts (as rural/urban areas or high/low-income users; e.g., [5,48]). In order to enforce this approach, future studies could consider a wider set of elements that shape the spatial, social, and regulative context of analysis. As regards social groups, for example, many studies assessing the impacts of AVs for users with high and low income mainly focus on the available budget. However, [34] highlight that many other elements would be worth considering. For instance, Americans with low income tend to have unstable car ownership, no bank account, live in suburbs poorly served by PT, and are less likely to work a nine-to-five job. All these aspects influence their accessibility. Therefore, the inclusion of these factors in accessibility models would make the analysis of social equity implications of AVs more accurate. However, this would also require the collection of detailed data.

(B) Approaches for accessibility analysis:

- Using statistical distribution measures to analyze the distribution effects of accessibility. To evaluate the implications of accessibility distribution effects for transport equity systematically, various studies have adopted statistical distribution measures like the Gini Index, the Theil Index, and the coefficient of variations (see the overview by [27]). In particular, the Gini Index is by far the most frequently used index to perform such distribution analysis because of its interpretability and ease of communication. In the literature discussing the impacts of AVs, the usage of this kind of tool would represent a novelty, which could support a more structured analysis of the transport equity implications of AVs.

- Considering different ethical stances to discuss transport equity implications. To evaluate the transport equity implications of AVs, an ethical problem is the definition of what is "fair". As suggested by [4], at least three theories on ethics are relevant for transport and accessibility evaluations: utilitarianism, egalitarianism, and sufficientarianism. Utilitarianism [70] aims to "maximize the gain", i.e., provide the highest benefit to the most significant part of the population. This theory is strongly linked to cost–benefit analysis, which is an integral approach in transport evaluation. Egalitarianism has a different perspective, since it aims to "minimize the pain". The theory of justice of [71] in particular argues that social goods labeled as "primary" should be guaranteed to all, and we should aim for the most significant benefit of the least advantaged members of society. Accordingly, it would be meaningful to focus on the accessibility gains of the social groups and areas experiencing the lowest accessibility level. Finally, sufficientarianism assumes that everyone should be well off. Therefore, there is a threshold defining what "sufficient" is, and our priority is to guarantee that everybody has a level of well-being over this threshold. In this case, the provision of a minimum accessibility level should be guaranteed to the whole society. In order to systematically discuss the potential impacts of AVs on transport equity, these three ethical stances could be considered in parallel and even compared.

- Analyzing accessibility and mobility implications jointly. As highlighted by [72], a sustainable transport planning approach is based on, among other things, a shift of perspective from mobility (the ability of people to move around) to accessibility (the ability of people to get what they need). According to this interpretation, accessibility and mobility are two very different but correlated concepts to take into account. For instance, accessibility can be increased without increasing mobility (e.g., by providing mixed functions at walkable distances). At the same time, increasing mobility could generate no benefit for accessibility (e.g., if the construction of a new highway lane generates enough induced demand). As such, future studies discussing the impacts of AVs should consider these two dimensions in parallel, in order to discuss the broader implications of AVs for the sustainability of the transport system. In this respect, various contributions suggest that AVs are generally likely to increase accessibility, but also mobility (e.g., [41]). Understanding how these two aspects could interfere with each other is a crucial target for future works.

Overall, it is important to mention that available literature discussing the accessibility impacts of AVs is still in its infancy (41 scientific contributions were identified). Therefore, these recommendations could be further developed in future in parallel with the development of this research field. Nevertheless, the information provided in this study and the suggestions included in this last section may be a basis for future works on the relation between AVs, accessibility, and transport equity. As highlighted, much can still be explored, both for the understanding of the assumptions that could play a key role, as well as for the analysis of the impacts.

**Author Contributions:** Conceptualization: A.D.; methodology: A.D.; validation: A.D., E.R. and G.H.; formal analysis: A.D.; investigation: A.D.; resources: A.D.; writing—original draft preparation: A.D.; writing—review and editing: A.D., E.R. and G.H.; visualization: A.D.; supervision: G.H.; funding acquisition: A.D. and E.R. All authors have read and agreed to the published version of the manuscript.

**Funding:** The APC was funded by the Department of Innovation, Research and University of the Autonomous Province of Bozen/Bolzano.

**Institutional Review Board Statement:** Not applicable.

**Informed Consent Statement:** Not applicable.

**Data Availability Statement:** No new data were created or analyzed in this study. Data sharing is not applicable to this article.

**Acknowledgments:** The authors thank the Department of Innovation, Research and University of the Autonomous Province of Bozen/Bolzano for covering the Open Access publication costs.

**Conflicts of Interest:** The authors declare no conflict of interest. The funders had no role in the design of the study; in the collection, analyses, or interpretation of data; in the writing of the manuscript; or in the decision to publish the results.

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
