# Peer review of "Implications of Autonomous Vehicles for Accessibility and Transport Equity: A Framework Based on Literature"

_sustainability, doi:10.3390/su13084448_

Round 1
Reviewer 1 Report
The article addresses how autonomous vehicles (AVs) impact accessibility and transport equity. Based on literature review as methodology, it provides a framework and discusses aspects related to the implications of AVs for the distribution of accessibility across space and social groups.
I find the paper interesting to read and that it is well written. The introduction explains why the topic is important to study and the gap in literature in an adequate way, and the aim of the paper is well explained.
The conceptual model is well described as well as the literature review, and the presentation and discussion of the results is well structured.
I only have one minor comment:
- Line 584: should be “shift” instead of “shit” at the start of the line
Author Response
Please see the attachment. In the file there is a point-by-point response to the comments provided by reviewers and and explanation of all applied revisions.

Reviewer 2 Report
The paper is well structured and grounded in literature.
It would be very helpful to have a section on the state of the art or a review of the literature highlighting some existing framework. This will place the work in a context.
The conclusion is also very long. Maybe you could have a discussion section before the conclusion and recommendation.
Author Response

(The authors gave the same response as above.)
